# INTROSPECTIVE ADVERSARIAL LEARNING: AUTONOMOUS AND CONTINUAL PREFERENCE LEARNING FOR LLM ALIGNMENT

## ABSTRACT

Large Language Models (LLMs) exhibit impressive capabilities across diverse tasks, yet aligning their outputs with human preferences remains a significant and costly challenge. Traditional alignment methods like Reinforcement Learning from Human Feedback (RLHF) depend heavily on extensive human-annotated preference data, which is difficult to scale. We propose Introspective Adversarial Learning (IAL), a novel alignment framework that enables LLMs to autonomously refine their own outputs through iterative self-improvement, without requiring additional human supervision. IAL introduces a Player-Advisor mechanism where the Player generates candidate responses and the Advisor provides constructive refinement strategies. The refined responses are evaluated by a reward model, and the contrast between initial and improved outputs drives a Preference Transductive Learning process. This reflective cycle allows the model to generate high-quality preference data internally and progressively enhance alignment. Experiments on the zephyr-7b-sft-full model, evaluated via the HuggingFace Open LLM Leaderboard and MT-Bench, show that IAL consistently improves alignment performance while preserving strong general task capabilities. Compared to state-of-the-art methods such as SPIN, SPA, and DPO, IAL achieves superior results without relying on costly human preference annotations, offering a scalable and efficient pathway toward better-aligned LLMs.

## 1 INTRODUCTION

Large language models (LLMs) (Liu et al., 2024; Ouyang et al., 2022; Achiam et al., 2023) have demonstrated remarkable capabilities in various domains such as Question Answering (Allam and Haggag, 2012; Zhang et al., 2023), Code Generation (Li et al., 2022; Svyatkovskiy et al., 2020), Summarization (Zhang et al., 2024; Pu et al., 2023), and Creative Writing. A significant advancement of these models is their astonishing problem-solving ability after training. However, this process relies heavily on costly human annotation efforts and consumes substantial computational resources. Moreover, ensuring that the responses generated by Large Language Models are aligned with human preferences remains a major challenge (Wang et al., 2023). The commonly used methods to enhance the alignment capabilities of Large Language Models are Supervised Fine-Tuning (SFT) (Ouyang et al., 2022; Tunstall et al., 2023) and Reinforcement Learning from Human Feedback (RLHF) (Christiano et al., 2017). In practice, both methods necessitate extensive and meticulously annotated datasets for model training. The acquisition of such data is often prohibitively expensive and time consuming. Consequently, researchers have increasingly focused on developing more efficient fine-tuning techniques that can maximize the utility of existing data to enhance alignment performance.

Recent advances have introduced self-enhancing algorithms that take advantage of the inherent generative capabilities of large models (Yuan et al., 2024; Wu et al., 2024). These algorithms are capable of transforming weakly aligned models into strongly aligned ones using minimal preference data or even only through fine-tuning data. Notable examples include Self-Play Fine-Tuning (SPIN) (Chen et al., 2024), Self-Play with Adversarial Critic (SPAC) (Ji et al., 2024), and Spread Preference Annotation (SPA) (Kim et al., 2025).

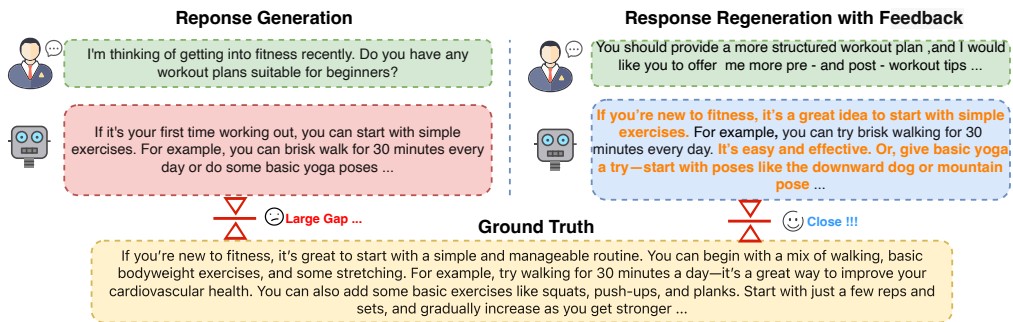

Figure 1: A comparative example of responses generated by the model with and without human feedback relative to ground truth. It can be observed that when the model receives human feedback, the response generated for the second time places greater emphasis on the detailed description of yoga movements and techniques compared to the initial response. The content of the response is also more closely aligned with the Ground Truth.

Despite these advancements, the alignment of large language models remains largely dependent on high-quality data manually annotated by humans (Tan et al., 2024; Chang et al., 2024). However, we have largely overlooked the potential of leveraging the models' inherent reasoning and summarization capabilities to generate higher-quality preference data pairs. Therefore, a crucial question arises: How can we effectively harness the intrinsic reasoning abilities of large language models to autonomously generate preference datasets and subsequently utilize these newly generated preference data to further enhance the alignment capabilities of the models?

In this paper, we introduce Introspective Adversarial Learning (IAL), an alignment framework that exploits the generative capacity of large language models to achieve strong alignment through fine-tuning on a dataset of aligned examples. The core idea is to iteratively engage the LLM in reinforcement learning–based adversarial training between self-generated and human-annotated data, thereby extracting improvement signals from their discrepancies. As illustrated in Figure 1, given the strong instruction-following abilities of LLMs, humans can provide prompts and suggestions that guide responses toward human preferences. In IAL, this suggestion role is assumed by the LLM itself. The model alternates between two roles—Player and Advisor—both initialized from the same base model. The Player executes complex instructions, while the Advisor offers constructive and realistic feedback on the Player's responses. The Player then refines its initial response by incorporating the Advisor's suggestions, producing a theoretically superior output. During training, both the initial and refined responses are collected, and their quality is evaluated and ranked using PairRM (Jiang et al., 2023), with the higher-scoring response designated as superior.

We conducted experiments on the Mistral-7b (Jiang, 2024) based fine-tuned model `zephyr-7b-sft-full` over multiple iterations. For training, we used the Ultrachat200k (Ding et al., 2023) dataset and evaluated the model performance on both the MT-Bench (Zheng et al., 2023) and the Open LLM Leaderboard (Beeching et al., 2023). Through these experiments, we demonstrated a significant enhancement in the model's alignment capabilities, outperforming baseline methods such as DPO (Rafailov et al., 2023), SPIN, and SPA. Importantly, our approach does not degrade the general benchmark performance of the LLM. In summary, our contributions are as follows:

1. **A Novel Alignment Framework**. We propose the Introspective Adversarial Learning (IAL) framework, which utilizes the generative capabilities of Large Language Models (LLMs) to autonomously produce high-quality preference data pairs via iterative adversarial training. This approach enhances model alignment without requiring additional human-annotated data, significantly reducing alignment costs.

2. **Innovative Player-Advisor Mechanism**. The IAL framework introduces a Player-Advisor mechanism. The Player executes complex instructions and generates initial responses, while

the Advisor provides optimization suggestions. This enables the model to self-assess and refine its responses, improving consistency with human preferences.

3. **Significant Improvement in Alignment Performance**. Experiments show that IAL significantly improves model alignment across multiple benchmarks, outperforming existing methods like SPIN, SPA, and DPO. Importantly, it maintains model performance on general tasks, demonstrating effective alignment without compromising generalization.

## 2 RELATED WORK

**Self-Play.** Self-Play (Samuel, 1959; Tesauro et al., 1995; Christiano et al., 2017; Silver et al., 2018) is a learning paradigm where an agent iteratively competes with itself. Widely used in Multi-Agent Reinforcement Learning (MARL) (Canese et al., 2021; Zhang et al., 2021; Wen et al., 2022), it pits the current model against its past iterations. Applications include Generative Adversarial Networks (GANs) (Goodfellow et al., 2020), adversarial imitation learning (Ho and Ermon, 2016), and inverse reinforcement learning (Ng and Russell, 2000). A prominent industrial example is AlphaGo Zero (Silver et al., 2017), which achieved superhuman performance via Self-Play.

In Large Language Model alignment, Self-Play Fine-Tuning (SPIN) employs Self-Play using a supervised fine-tuning dataset $(x, y)$ to generate responses $y'$. Without strong model supervision, it iteratively refines the model to better distinguish $y$ from $y'$, producing a stronger model. Unlike SPIN, Self-Play with Adversarial Critic (SPAC) uses a preference dataset. This offline preference optimization method frames the problem as a Stackelberg game (Nie and Zhang, 2008), introducing an adversarial critic to maintain a pessimistic reward estimate while optimizing the policy.

**Iterative Self-Improvement in LLMs.** Growing interest exists in methods enabling LLMs to improve autonomously through iterative self-training. Self-Critique (Li, 2024) and Self-Rewarding (Yuan et al., 2024) let models generate and evaluate their own outputs. Coffee (Moon et al., 2023) uses self-generated feedback to enhance code generation. Similarly, Self-Play Preference Optimization (SPPO) (Wu et al., 2024) extends self-play to preference learning. The Sol-Ver Lin et al. (2025) framework enhances code and test generation through solver–verifier self-play, whereas the SPC Chen et al. (2025) framework employs adversarial self-play to refine the critic's reasoning evaluation, thereby strengthening LLM reasoning performance.

**Alignment of LLMs with Human Preferences.** Current alignment methods primarily use Reinforcement Learning from Human Feedback (RLHF), training a reward model on human preferences and optimizing the policy with algorithms like Proximal Policy Optimization (PPO) (Schulman et al., 2017), using KL divergence to avoid excessive deviation. Direct Preference Optimization (DPO) simplifies this by using an implicit reward function, bypassing explicit reward modeling. Advances like SimPO (Meng et al.) and DPO-Positive (Pal et al., 2024) have improved performance, yet these methods still rely on costly, time-consuming human-annotated datasets.

## 3 METHODOLOGY

**Overview**. In this section, we introduce a novel fine-tuning method for enhancing the alignment of Large Language Model (LLM) without the need for preference datasets or human feedback, termed Introspective Adversarial Learning. As illustrated in Figure 2, our proposed method consists of two main components: Self-Refined Responses Generation and Preference Transductive Learning. By leveraging offline data generation and reinforcement learning, we can transform a weakly aligned Large Language Model into a strongly aligned LLM.

### 3.1 BACKGROUND

We consider a Large Language Model parameterized by parameters and denoted as $\pi_\theta$. In the context of preference learning, given a text sequence $\mathbf{x} = [x_1, x_2, \ldots, x_n]$, the model generates a response $\mathbf{y} = [y_1, y_2, \ldots, y_m]$ relative to $\mathbf{x}$. The response $\mathbf{y}$ can be regarded as a sample drawn from $\pi_\theta(\cdot|\mathbf{x})$. It is worth noting that $x_i$ and $y_i$ represent the individual tokens at the $i$-th position in the sequences $\mathbf{x}$ and $\mathbf{y}$, respectively. The process of generating responses by an autoregressive model is a Markov process (Ethier and Kurtz, 2009). The autoregressive model $\pi_\theta$ utilizes the previously generated

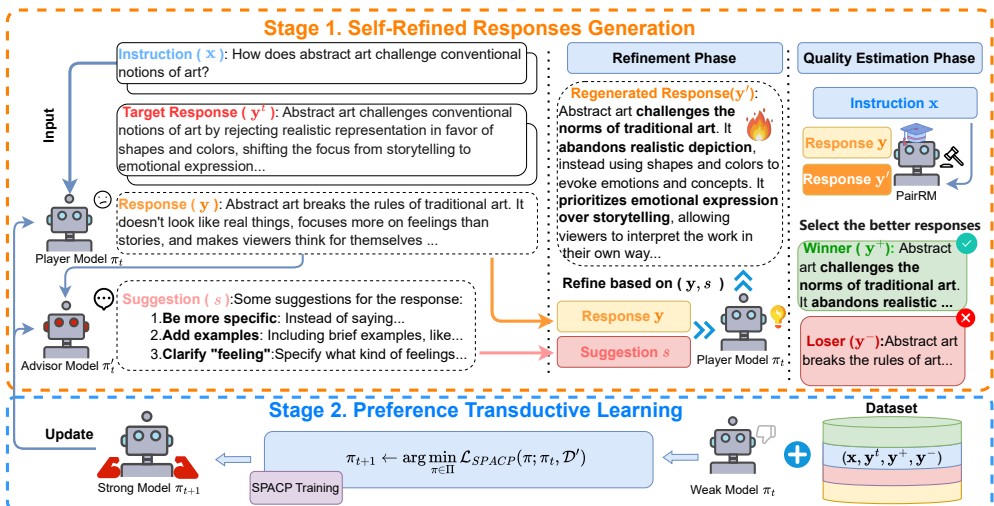

Figure 2: The framework of Introspective Adversarial Learning ( IAL). In the $t$-th iteration of IAL, the Player model samples an initial Response $\mathbf{y}$ from the Instruction $\mathbf{x}$, and the Advisor model generates a Suggestion $s$ based on $\mathbf{x}$ and $\mathbf{y}$. The Player model then produces a Regenerated Response $\mathbf{y}'$ using $s$ and ranks it against $\mathbf{y}$. SPACP optimizes the model in the Preference Transductive Learning phase for continuous self-improvement.

token sequence to generate a token for a given position in the sequence. Given the prompt $\mathbf{x}$, the autoregressive language model $\pi_\theta$ can generate the response $\mathbf{y}$ through probabilistic means:

$$\pi_\theta(\mathbf{y}|\mathbf{x}) = \prod_{j=1}^{m} \pi_\theta(y_j|\mathbf{x}, \mathbf{y}_{<j}) \tag{1}$$

Given a human preference dataset $D = \{\mathbf{x}_j, \mathbf{y}_j^+, \mathbf{y}_j^-\}_{j=1}^{N}$, where $\mathbf{x}_j$ denotes the $j$-th instruction in the dataset, commonly referred to as a Prompt, and $\mathbf{y}_j^+$ and $\mathbf{y}_j^-$ represent the preferred and less preferred responses, respectively, conditioned on the given instruction $\mathbf{x}_j$. Assuming the existence of a model $\pi_{\text{ref}}$ that has been fine-tuned with supervised learning, the Reinforcement Learning from Human Feedback (RLHF) method employs the Bradley-Terry (David, 1963) model to characterize the aforementioned preference responses. Specifically, the probability of the preferred response emerging victorious is modeled as follows:

$$p(\mathbf{y}^+ \succ \mathbf{y}^-|\mathbf{x}) = \sigma(r(\mathbf{x}, \mathbf{y}^+) - r(\mathbf{x}, \mathbf{y}^-)) \tag{2}$$

where $\sigma$ represents the logistic function, and $r(x, y)$ denotes a latent reward function that is hypothesized to exist for the response $y$ given the prompt $x$. In the context of dataset $\mathcal{D}$, the training objective of the reward model is to encourage the model to assign higher scores to preferred responses than to dispreferred ones. Formally, the loss function is defined as:

$$\mathcal{L}_{\text{RM}}(r, \mathcal{D}) = -\mathbb{E}_{(\mathbf{x}, \mathbf{y}^+, \mathbf{y}^-) \sim \mathcal{D}} \Big[ \log \sigma \big( r(\mathbf{x}, \mathbf{y}^+) - r(\mathbf{x}, \mathbf{y}^-) \big) \Big]. \tag{3}$$

## 3.2 SELF-REFINED RESPONSES GENERATION

**Initiation**. We assume that we are given a large fine-tuning dataset $\mathbf{D_L}$ and an initial parameterization $\pi_{\text{init}}$ of the Large Language Model. The LLM is required to undergo standard Supervised Fine-Tuning (SFT) on the dataset $\mathbf{D_L}$ prior to the subsequent procedures. The loss function for Supervised Fine-Tuning is defined as follows:

$$\mathcal{L}_{SFT}(\pi_\theta, \mathbf{x}, \mathbf{y}) = -\frac{1}{N} \sum_{i=1}^{N} \log \pi_\theta(y_i|\mathbf{x}, \mathbf{y}_{<i}) \tag{4}$$

In this context, $\mathbf{x}$ signifies the given fine-tuning prompt, and $N$ denotes the length of the entire fine-tuning data response. Post fine-tuning, the model, denoted as $\pi_0$, is a weakly aligned model and has not yet been aligned with human preferences.

In the $j$-th iteration, a small fine-tuning dataset $D_j \sim \mathbf{D_L}$ is randomly sampled from the large fine-tuning dataset $\mathbf{D_L}$, where $D_j = \{(\mathbf{x}_i, \mathbf{y}_i^t) \mid 0 \leq i \leq N - 1\}$. It is noteworthy that $\mathbf{y}_i^t$ denotes the target response in our fine-tuning dataset. We consider a scenario in which humans interact with a Large Language Model through dialogues. As shown in Figure 1, in this interaction, a specific task prompt $\mathbf{x}_i$ is provided to the LLM, which then generates a corresponding response $\mathbf{y}_i$. Based on the task requirements, humans provide textual feedback and Suggestion $s_i$ for the LLM's response, such as making the response more conversational, reducing the length of the response, or providing a more detailed explanation of the answer. Based on these suggestions, the LLM can generate a new response $\mathbf{y}_i'$ that is theoretically more aligned with the task requirements and human preferences.

In this process, the LLM acts as a **Player**, generating responses based on the given instructions, while humans act as **Advisors**, not only evaluating the LLM's responses but also providing directions for optimization. The complex reasoning and response optimization suggestions provided by humans can be emulated by a single fine-tuned LLM. The entire dialogue process can be realized based on an initialized LLM model parameter $\pi_t$. However, it is important to recognize that there may be discrepancies between the preferences of the unaligned LLM and human preferences. The regenerated response $\mathbf{y}_i'$ may not necessarily be more aligned with human preferences compared to the initially sampled response $\mathbf{y}_i$. Therefore, an additional reward model $r_\theta$ is required to evaluate preferences and select responses that are more aligned with human preferences $(\mathbf{y}_i^+)$ and those that are not recommended $(\mathbf{y}_i^-)$.

$$(\mathbf{y}_i^+, \mathbf{y}_i^-) = (\mathbf{y}_i, \mathbf{y}_i') \text{ if } r_\theta(\mathbf{y}_i) > r_\theta(\mathbf{y}_i') \text{ else } (\mathbf{y}_i', \mathbf{y}_i) \tag{5}$$

Here, $r_\theta(\mathbf{y})$ denotes the evaluation score assigned to the response $\mathbf{y}$ by the reward model, where a higher score indicates greater alignment with the preferences of the reward model. Based on the aforementioned procedures, upon the completion of the $j$-th iteration of data generation, we construct a preference dataset $D_j'$ for subsequent model training, where $(\mathbf{x}, \mathbf{y}^t, \mathbf{y}^+, \mathbf{y}^-) \in D_j'$.

### 3.3 PREFERENCE TRANSDUCTIVE LEARNING

We formulate the offline preference optimization as a Stackelberg game between two players: a Policy (Leader) and an Adversarial Critic (Follower) . The policy $\pi_\theta$ aims to maximize the value estimated by the critic, while the critic constructs a pessimistic reward estimate constrained by preference data. This duality is achieved through a bilevel optimization framework.

SPAC (Ji et al., 2024) has already specified the objective of the critic, which is to minimize a loss function that integrates the reward estimate and the average pessimism for a fixed policy $\pi_\theta$, where $D = \{(\mathbf{x}_i, \mathbf{y}_i^+, \mathbf{y}_i^-) \mid 0 \leq i \leq N - 1\}$:

$$\min_r \underbrace{\mathbb{E}_{(\mathbf{x},\mathbf{y}^+,\mathbf{y}^-)\sim\mathcal{D}} \left[ -\log \sigma \left( r(\mathbf{x}, \mathbf{y}^+) - r(\mathbf{x}, \mathbf{y}^-) \right) \right]}_{\text{Preference Loss}} + \lambda \underbrace{\mathbb{E}_{\mathbf{x}\sim\mathcal{D}, \mathbf{y}\sim\pi_\theta(\cdot|\mathbf{x})} \left[ r(\mathbf{x}, \mathbf{y}) \right]}_{\text{Pessimism Penalty}} \tag{6}$$

where $\lambda > 0$ controls the strength of pessimism. In our policy update phase, the policy $\pi_\theta$ is updated to maximize the critic's pessimistic reward estimate while maintaining proximity to the reference model $\pi_{\text{ref}}$:

$$\max_{\pi_\theta} \mathbb{E}_{\mathbf{x}\sim\mathcal{D}, \mathbf{y}\sim\pi_\theta(\cdot|\mathbf{x})} \left[ r(\mathbf{x}, \mathbf{y}) - \alpha D_{\text{KL}} \left( \pi_\theta(\cdot|\mathbf{x}) \big\| \pi_{\text{ref}}(\cdot|\mathbf{x}) \right) \right] \tag{7}$$

It is worth mentioning that, to avoid explicit reward modeling, we employ a DPO-inspired transformation. Let $r(\mathbf{x}, \mathbf{y}) = \eta \log \frac{\pi_\theta(\mathbf{y}|\mathbf{x})}{\pi_{\text{ref}}(\mathbf{y}|\mathbf{x})} + \log Z(\mathbf{x})$, where $Z(\mathbf{x})$ is a normalization term. Substituting this into the critic's objective yields a practical single-timescale algorithm:

$$\min_{\pi_\theta} -\mathbb{E}_{(\mathbf{x},\mathbf{y}^+,\mathbf{y}^-)\sim\mathcal{D}} \log \sigma \left( \eta \log \frac{\pi_\theta(\mathbf{y}^+|\mathbf{x})}{\pi_{\text{ref}}(\mathbf{y}^+|\mathbf{x})} - \eta \log \frac{\pi_\theta(\mathbf{y}^-|\mathbf{x})}{\pi_{\text{ref}}(\mathbf{y}^-|\mathbf{x})} \right)$$
$$+ \lambda \cdot \mathbb{E}_{\mathbf{x}\sim\mathcal{D}, \mathbf{y}\sim\pi_\theta(\cdot|\mathbf{x})} \log \sigma \left( \eta \log \frac{\pi_\theta(\mathbf{y}|\mathbf{x})}{\pi_{\text{ref}}(\mathbf{y}|\mathbf{x})} \right) \tag{8}$$

where $\eta$ governs the alignment between the reward and the policy. After minor enhancements, the resulting formula is presented as follows:

$$\pi_{t+1} \leftarrow \arg\min_{\pi \in \Pi} \mathbb{E}_{(\mathbf{x},\mathbf{y}^+,\mathbf{y}^-)\sim\mathcal{D},\mathbf{y}\sim\pi_\theta(\cdot|\mathbf{x})} \eta \left( \log\frac{\pi(\mathbf{y}|\mathbf{x})}{\pi_t(\mathbf{y}|\mathbf{x})} - \log\frac{\pi(\mathbf{y}^+|\mathbf{x})}{\pi_t(\mathbf{y}^+|\mathbf{x})} \right)$$

$$-\lambda \mathbb{E}_{(\mathbf{x},\mathbf{y}^+,\mathbf{y}^-)\sim\mathcal{D}} \log\sigma \left( \eta\log\frac{\pi(\mathbf{y}^+|\mathbf{x})}{\pi_t(\mathbf{y}^+|\mathbf{x})} - \eta\log\frac{\pi(\mathbf{y}^-|\mathbf{x})}{\pi_t(\mathbf{y}^-|\mathbf{x})} \right) \quad (9)$$

where, $\pi_t$ represents the model at t-th iteration, $\mathcal{D}$ represents the preference dataset, and $\Pi$ signifies the policy class.

However, the entire approach is fundamentally grounded in the Bradley–Terry (BT) (Bradley and Terry, 1952) model. In Direct Preference Optimization (DPO) (Rafailov et al., 2023), the BT formulation only enforces relative ranking constraints, without directly guaranteeing an increase in the probability of positive samples. This introduces a potential weakness (Pal et al., 2024; Ren and Sutherland, 2024): *during training, the model may sacrifice the log-probability of $\mathbf{y}^+$ while achieving the optimization objective primarily through more aggressively suppressing $\mathbf{y}^-$.*

When optimizing according to Equation 9, this issue inevitably arises in practice. We provide a more detailed empirical investigation of this phenomenon in the Appendix B.3.

To address the aforementioned issue, we propose **Self-Play with Adversarial Critic-Positive (SPACP)**, a novel reinforcement learning strategy. Given the high confidence in the target responses within the fine-tuning dataset, we incorporate the penalty term $P_{\pi,\pi_t}(\mathbf{x},\mathbf{y}^t) = \max\left(0, \log\frac{\pi_t(\mathbf{y}^t|\mathbf{x})}{\pi(\mathbf{y}^t|\mathbf{x})}\right)$ into the log-sigmoid loss function of Equation 9. The integration of this penalty term is intended to preserve a high log-likelihood for the target responses. The complete loss function for SPACP is as follows, where $D' = \{(\mathbf{x}_i, \mathbf{y}_i^t, \mathbf{y}_i^+, \mathbf{y}_i^-) \mid 0 \le i \le N-1\}$, $\gamma$ is a hyperparameter used to control the strength of the penalty term:

$$\mathcal{L}_{SPACP}(\pi;\pi_t,\mathcal{D}') = \mathbb{E}_{(\mathbf{x},\mathbf{y}^t,\mathbf{y}^+,\mathbf{y}^-)\sim\mathcal{D}'} \eta \left( \log\frac{\pi(\mathbf{y}^+|\mathbf{x})}{\pi_t(\mathbf{y}^+|\mathbf{x})} - \log\frac{\pi(\mathbf{y}^t|\mathbf{x})}{\pi_t(\mathbf{y}^t|\mathbf{x})} + \gamma \cdot P_{\pi,\pi_t}(\mathbf{x},\mathbf{y}^t) \right)$$

$$-\lambda \mathbb{E}_{(\mathbf{x},\mathbf{y^t},\mathbf{y}^+,\mathbf{y}^-)\sim\mathcal{D}'} \log\sigma \left( \eta\log\frac{\pi(\mathbf{y}^t|\mathbf{x})}{\pi_t(\mathbf{y}^t|\mathbf{x})} - \eta\log\frac{\pi(\mathbf{y}^-|\mathbf{x})}{\pi_t(\mathbf{y}^-|\mathbf{x})} - \eta\cdot\gamma\cdot P_{\pi,\pi_t}(\mathbf{x},\mathbf{y}^t) \right)$$

$$(10)$$

By incorporating a penalty term, we can enhance the log-likelihood of the target responses within the fine-tuning dataset. This approach not only increases the gap in the log-probabilities between the target and non-target responses but also ensures that the log-likelihood of the target responses is improved compared to the model from the previous iteration. This method effectively guides the model alignment in a deterministic direction during the Self-Play process, thereby preventing the accumulation of alignment errors.

After each training session, the model parameters of both the Player and Advisor are updated using the most recent parameters, thereby commencing the next iteration. By integrating self-refinement of data generation with the Self-Play training process, IAL enhances the alignment of weakly aligned models, ultimately transforming them into strongly aligned models. This approach leverages iterative refinement to strengthen model alignment, thereby ensuring more robust and accurate model performance.

## 4 EXPERIMENTS

In this section, we conduct a detailed experimental analysis of IAL. Through our experiments, we demonstrate that IAL can outperform SPIN, SPA, and DPO (which requires additional human-annotated data) without the need for extra human-labeled data. This highlights that IAL achieves superior performance across various evaluation benchmarks compared to the baseline.

### 4.1 EXPERIMENT SETUP

**Model and Datasets**. In our research, we employ the `zephyr-7B-sft-full` as the foundational architecture. This model is derived from the pre-trained Mistral-7B (Jiang, 2024) and has been fine-tuned on the SFT dataset Ultrachat200k from HuggingFace. The Ultrachat200k dataset represents a

high-quality subset of 200,000 samples from the larger UltraChat (Ding et al., 2023) corpus, which includes approximately 1.4 million dialogues generated using OpenAI's Turbo API. To facilitate comparisons with DPO (Direct Preference Optimization) and SPA (Spread Preference Annotation), we utilize the widely adopted UltraFeedback (Cui et al., 2023) dataset from prior work (Hong et al., 2024; Rosset et al., 2024) for training these methods. For the external reward model, we employ PairRM (Jiang et al., 2023) for preference judgment. Drawing inspiration from the data processing methodology of the SPIN (Chen et al., 2024) dataset, we initiate our iterative process by randomly sampling 50,000 data points from Ultrachat200k for data generation. Subsequently, we train the model according to the algorithm described in Section 3.3 of this paper. In the ensuing iterative training phases, we blend the most recently synthesized data with newly generated data, training the model for two epochs in each iteration. For additional experiments with other model, please refer to the Appendix B.2.1.

**Evaluation**. In this study, we adopt two widely recognized benchmarks for assessing the performance of Large Language Models (LLMs): the Huggingface Open LLM Leaderboard (Beeching et al., 2023) and MT-Bench (Zheng et al., 2023). These benchmarks are commonly utilized within the research community to evaluate various dimensions of LLM capabilities. (1) The Huggingface Open LLM Leaderboard consists of six diverse datasets that provide a comprehensive evaluation of models from multiple angles. It covers a wide range of Natural Language Processing (NLP) tasks, including but not limited to Multitask Language Understanding (MMLU), Commonsense Reasoning (Arc (Clark et al., 2018), HellaSwag (Zellers et al., 2019), Winogrande (Sakaguchi et al., 2019)), Human Deception detection (TruthfulQA) (Lin et al., 2021), and Mathematical Inference (GSM8K (Cobbe et al., 2021)). These tasks span multiple domains, allowing for a robust and holistic evaluation of the LLM's generalization capabilities. (2) MT-Bench, on the other hand, specifically evaluates the overall performance of chatbots across several key categories that are critical for LLM proficiency, such as mathematical reasoning, programming, role-playing, and creative writing. The evaluation methodology involves scoring multi-turn responses generated by the models, with GPT-4 being employed to assess the quality of these responses. For a more detailed discussion of the evaluation methodology, please refer to the Appendix B.1.

**Implementation Details**. In each iteration, we sample a seed dataset from the fine-tuning dataset. Utilizing the prompts within this seed dataset, we perform Self-Refined Responses Generation. For each prompt, we sample a single response with a temperature setting of 0.7. Subsequently, we assign preference labels to the generated response pairs using PairRM (Jiang et al., 2023). The entire iterative process is conducted for three iterations, with each iteration lasting for two epochs. For the hyperparameters in IAL, we set $\beta = 0.1$, $\gamma = 5.0$, and $\lambda = 1.0$, with an initial learning rate of $5 \times 10^{-7}$. We utilize the RMSProp optimizer and a linear learning rate scheduler, where the warm-up phase constitutes 10% of the total training steps.

## 4.2 BASELINES SETTINGS

In this section, we will provide a detailed description of the training details of the baselines we employed, as well as the datasets utilized. For additional experiments with more baselines, please refer to the Appendix B.2.2.

**DPO**. We conducted three iterations of training using the `zephyr-7B-sft-full` model on the UltraFeedback dataset, with each iteration comprising two epochs. The warm-up steps accounted for 10% of the total training steps. We employed the AdamW optimizer with a learning rate of $5.0 \times 10^{-7}$ and a hyperparameter $\beta = 0.1$. Additionally, we utilized a Cosine learning rate scheduler. These hyperparameters are widely adopted, and their training outcomes can be regarded as the ideal reference results for DPO.

**SPA**. After initializing with DPO, we sampled each independent prompt twice from the seed dataset with a temperature of 0.7 and assigned preference labels. For the DPO hyperparameter $\beta$, we maintained a fixed value of $\beta = 0.1$ and a learning rate of $5 \times 10^{-7}$. We adopted the AdamW optimizer and a Cosine learning rate scheduler, with the warm-up phase corresponding to 10% of the total training steps. For the SPA hyperparameters $\alpha$ and $K\%$, we used fixed values of $\alpha = 0.1$ and $K = 10$, and we performed two iterations in total.

**SPIN**. In each iteration, we randomly sampled 50k prompts from the UltraChat200k dataset and generated synthetic responses using the base model. Following the SPIN experimental configuration,

we utilized the synthetic data from the most recent iteration and added it to the newly generated synthetic data. Consequently, the synthetic dataset size was 50k in iteration 0, and 100k in iterations 1 and 2. In each iteration, we trained the model for two epochs. We selected $\beta = 0.1$ and the RMSProp optimizer, with warm-up steps accounting for 10% of the total training steps. For iterations 0 and 1, the learning rate was set to $5 \times 10^{-7}$, while for iteration 2, it was set to $1 \times 10^{-7}$.

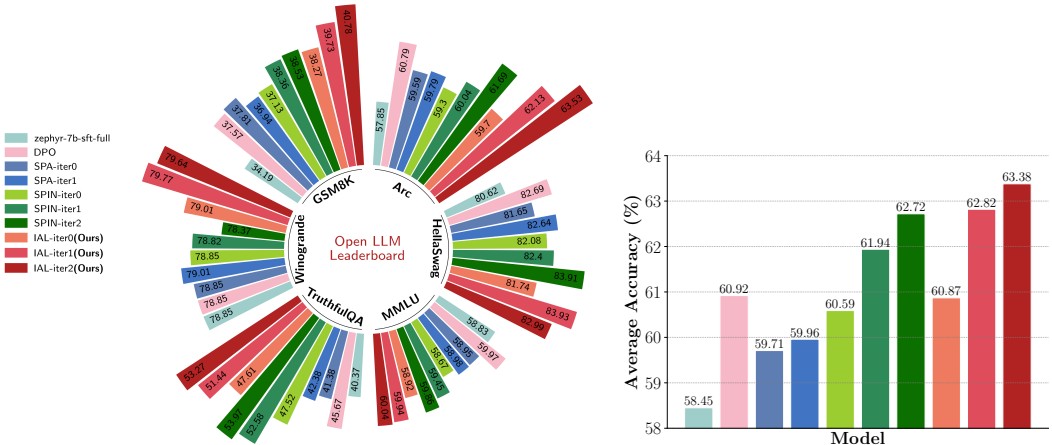

Figure 3: The evaluation results of baselines and IAL on the HuggingFace Open LLM Leaderboard. See Table 2 for detailed data.

## 4.3 MAIN RESULTS

Based on the results shown in Figure 3, we utilized the HuggingFace Open LLM Leaderboard (Beeching et al., 2023) as our evaluation benchmark to demonstrate the superiority and effectiveness of our method. We compared the performance of models trained with our approach against those trained with SPA (Kim et al., 2025), SPIN (Chen et al., 2024), and DPO (Rafailov et al., 2023) methods. We specifically examined the differences in evaluation outcomes between models after three iterations of IAL and those after initial full fine-tuning, significantly highlighting our method's capability to enhance model alignment. At the initial iteration, we used the fine-tuned zephyr-7b-sft-full to generate a dataset for subsequent model training. Given that the model was still in a weak alignment state, we observed an average evaluation accuracy increase of 2.42% after the first iteration, primarily improving in the TruthfulQA (Lin et al., 2021), GSM8K (Cobbe et al., 2021), and HellaSwag (Zellers et al., 2019) metrics. During the second iteration, the model continued to enhance all metrics, with the HellaSwag metric surpassing the maximum values of other baselines. The average evaluation accuracy also exceeded the maximum values of other baselines, improving by 1.95% compared to the previous iteration. In the final iteration, although the model showed a decrease in Winogrande and HellaSwag metrics, other metrics increased. Ultimately, our method achieved an average evaluation accuracy of 63.38%, a 0.42% improvement over the previous iteration, surpassing the three baseline algorithms. It can be observed that as iterations progressed, our model gradually transitioned from weak to strong alignment, with diminishing improvement margins, yet still significantly outperforming DPO, which relies on additional preference data for training.

We conducted additional evaluations on the MT-Bench (Zheng et al., 2023), where we compared the optimal results obtained from multiple iterations of various baseline models with the final model trained using the Iterative Active Learning (IAL) approach, as illustrated in Figure 4. Upon a thorough analysis, it is evident that the IAL-trained model shows significant improvements over the baseline model, which was trained using Supervised Fine-Tuning (SFT), across several key metrics on the MT-Bench. Notably, the most substantial enhancement is observed in the Writing metric, where the IAL model outperforms its counterparts by a considerable margin. Furthermore, IAL also demonstrates robust performance in the Roleplay metric, highlighting its versatility across different task categories. The final model, trained with IAL, achieved an impressive average score of 6.96, marking an enhancement of 0.98 points over the SFT-trained model and surpassing the performance

of other baseline models. For a detailed breakdown of the experimental results and further analysis, please refer to the Appendix B.2.

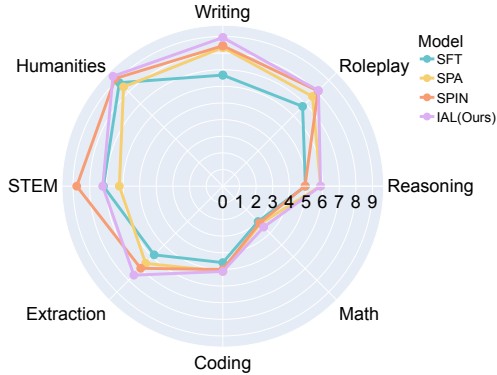 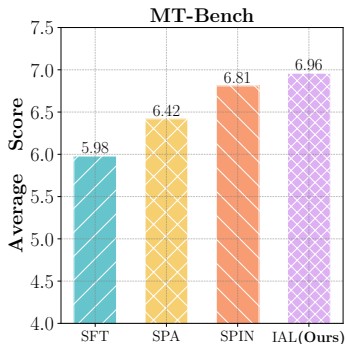

Figure 4: **Left**. Comparison of the best evaluation results across baselines on MT-Bench. **Right**. Average scores, demonstrating the superiority of our method over the baselines.

## 4.4 ABLATION STUDIES

In this subsection, we examine the effect of the number of training epochs and the hyperparameter $\gamma$ on model training. Starting from the model after initial Supervised Fine-Tuning (SFT), we experiment with different $\gamma$ values and evaluate performance on the HuggingFace Open LLM Leaderboard. This allows us to analyze how $\gamma$ influences model alignment. Figure 5 shows the performance of the initial model under different $\gamma$ values and training iterations. For $\gamma$, average accuracy on the HuggingFace benchmark rises steadily early on, with the largest gain in the first two epochs. In contrast, for $\gamma = 1$, performance improves more slowly, only surpassing the degraded $\gamma = 5$ case by the fifth epoch. For $\gamma = 50$ and $\gamma = 500$, performance declines as training progresses, likely because high $\gamma$ makes the process similar to SFT. Since the model is already fine-tuned on this dataset, further SFT causes overfitting and performance drop. Thus, proper selection of $\gamma$ and epoch count is essential to improve model robustness and accuracy. Additional ablation studies are provided in Appendix B.4.

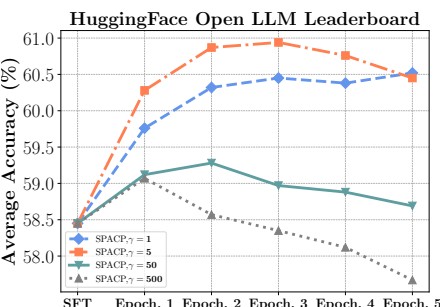

Figure 5: The models trained with different $\gamma$ values for SPACP were evaluated on the Hugging Face Open LLM Leaderboard, demonstrating the trend of average accuracy.

## 5 CONCLUSIONS

This paper introduces Introspective Adversarial Learning (IAL), an alignment framework that harnesses the generative power of Large Language Models (LLMs) to autonomously produce high-quality preference data, thereby improving model alignment. Unlike traditional approaches, IAL eliminates the need for extra human-annotated data, reducing cost and increasing efficiency. Extensive experiments on the HuggingFace Open LLM Leaderboard and MT-Bench show that IAL outperforms state-of-the-art methods such as SPIN, SPA, and DPO in alignment capability. Moreover, IAL maintains strong model performance on benchmark tasks, balancing alignment gains with overall competence.

Future work will extend IAL to diverse LLMs and tasks to assess its generality, while also aiming to enhance alignment performance and streamline training for greater efficiency. We believe IAL offers valuable insights and directions for LLM alignment research.

## Ethics Statement

We hereby declare that the present study, which proposes and validates Introspective Adversarial Learning (IAL) for enhancing the alignment of large language models (LLMs) with human preferences while reducing reliance on human-annotated data, strictly adheres to academic ethics in its design, experimentation, and documentation. All datasets used in this study (e.g., UltraFeedback) are publicly available academic resources containing no personally identifiable information or sensitive data. All data processing procedures comply with the terms of use provided by the data sources, and no illegal scraping or unauthorized use of data was conducted.

Although this research employs LLMs to generate textual content (e.g., suggestions, responses), all generated materials have been manually reviewed and verified to ensure compliance with academic integrity and social ethics. We commit not to produce or disseminate any misleading, harmful, or unethical content.

We affirm that the goal of this study is to advance the development of AI alignment techniques, thereby contributing to the creation of safer, more reliable, and human-value-aligned AI systems. We welcome the research community to examine, verify, and critique our methodology, experiments, and conclusions.

## Reproducibility Statement

To ensure the reproducibility of our work, we have made our code publicly available in an anonymous repository: https://anonymous.4open.science/r/IAL-27E3. The experimental environment can be installed using the provided requirements.txt file. As detailed in Section 4.1 of the paper, we specify the models and open-source datasets used in our experiments. Furthermore, Table B.1 describes the evaluation metrics and settings in detail. Additional implementation specifics, including the open-source libraries utilized and the dialogue templates for LLM interactions, can be found in the anonymous code repository.

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

APPENDIX

## A    THE USE OF LARGE LANGUAGE MODELS

In the preparation of this manuscript, including the appendix, we employed Large Language Models (LLMs) for grammar checking and refinement of academic writing style to enhance readability and professionalism. In addition, we utilized Artificial Intelligence (AI) to convert handwritten mathematical formula images into LaTeX code, thereby facilitating the integration of equations into the paper. All AI-generated content was carefully reviewed and verified by the authors, and we assume full responsibility for the use of generative AI in this work.

## B    FURTHER DETAILS ON THE EXPERIMENT

### B.1    OTHER EXPERIMENTAL SETTINGS AND EVALUATION DETAILS

Due to limited computational resources, we utilized Lora and DeepSpeed ZeRO-3 throughout the experiments, with a global batch size of 4, a Gradient Accumulation size of 8, and precision set to bfloat16.

Table 1: Detailed information of HuggingFace Open LLM Leaderboard.

| Datasets | Arc | TruthfulQA | Winogrande | GSM8k | HellaSwag | MMLU |
|---|---|---|---|---|---|---|
| # few-shot | 25 | 0 | 5 | 5 | 10 | 5 |
| Metric | acc_norm | mc2 | acc | acc | acc_norm | acc |

Table 2: Compared to all baselines, the IAL based on `zephyr-7B-sft-full` demonstrates superior performance in the HuggingFace Open LLM Leaderboard dataset. The best results in each column are highlighted in **bold**. The second-best results are highlighted with an underline. (we have also added the increment compared to the previous iteration in the Average column).

| Model | Arc | HellaSwag | MMLU | TruthfulQA | Winogrande | GSM8K | **Average** |
|---|---|---|---|---|---|---|---|
| `zephyr-7b-sft-full` | 57.85 | 80.62 | 58.83 | 40.37 | 78.85 | 34.19 | 58.45 |
| `DPO` | 60.79 | 82.69 | 59.97 | 45.67 | 78.85 | 37.57 | 60.92(+2.47) |
| `SPA-iter0` | 59.59 | 81.65 | 58.95 | 41.38 | 78.85 | 37.81 | 59.71(+1.26) |
| `SPA-iter1` | 59.79 | 82.64 | 58.98 | 42.38 | 79.01 | 36.94 | 59.96(+0.25) |
| `SPIN-iter0` | 59.30 | 82.08 | 58.67 | 47.52 | 78.85 | 37.13 | 60.59(+2.14) |
| `SPIN-iter1` | 60.04 | 82.40 | 59.45 | 52.58 | 78.82 | 38.36 | 61.94(+1.35) |
| `SPIN-iter2` | 61.69 | 83.91 | 59.86 | **53.97** | 78.37 | 38.53 | 62.72 (+0.78) |
| `IAL-iter0`**(Ours)** | 59.70 | 81.74 | 58.92 | 47.61 | 79.01 | 38.27 | 60.87(+2.42) |
| `IAL-iter1`**(Ours)** | 62.13 | **83.93** | 59.94 | 51.44 | **79.77** | 39.73 | 62.82(+1.95) |
| `IAL-iter2`**(Ours)** | **63.53** | 82.99 | **60.04** | 53.27 | 79.64 | **40.78** | **63.38**(+0.42) |

### B.2    MORE EXPERIMENT RESULT

#### B.2.1    OUR METHOD ON QWEN-2.5

Evaluating our method on a single model is insufficient to fully demonstrate its effectiveness. To provide a more comprehensive assessment, we conducted additional experiments on `Qwen-2.5-3B` (Hui et al., 2024). Qwen-2.5, a member of the Qwen large language model series, features a substantially enlarged knowledge base and exhibits marked improvements in programming and mathematical reasoning capabilities. As reported in Table 2, the SPA method demonstrates comparatively weak performance; therefore, we focus our experiments on DPO, SPIN, and IAL (our proposed method). For fair comparison and reproducibility, the training hyperparameters are kept consistent with those described in Section 4.2. Table 3 summarizes the evaluation results of these methods on the Open LLM Leaderboard.

Table 3: Evaluation results of `Qwen-2.5-3B` and its fine-tuned variants on the Open LLM Leaderboard. The best results in each column are highlighted in **bold**. The second-best results are highlighted with an underline.

| Model | Arc | HellaSwag | MMLU | TruthfulQA | Winogrande | GSM8K | Average |
|---|---|---|---|---|---|---|---|
| `Qwen-2.5-3B` | 57.08 | 74.45 | 65.62 | 48.96 | **71.27** | 75.51 | 65.48 |
| DPO | 58.70 | 78.32 | 65.87 | **56.93** | 68.67 | 78.32 | 67.31(+1.83) |
| SPIN | 59.40 | 78.26 | 66.12 | 55.63 | 68.89 | 78.51 | 67.80(+2.32) |
| IAL **(Ours)** | **59.89** | **79.34** | **66.97** | 56.54 | 68.74 | **78.62** | **68.35**(+3.26) |

### B.2.2 MORE BASELINES

We conducted a more thorough exploration and evaluation of the baseline methods, as detailed below:

**SFT&DPO**. Prior studies (Saeidi et al., 2024) have indicated that, when applied to relatively weaker base models, performing task-specific Supervised Fine-Tuning prior to Direct Preference Optimization is advantageous, leading to more stable and substantial performance improvements. We first conducted SFT on the `zephyr-7b-sft-full` model using the chosen responses from the UltraFeedback Binarized dataset prior to initiating DPO training. The SFT stage was performed for 2 epochs with a learning rate of $2 \times 10^{-4}$, employing the AdamW optimizer. Subsequently, DPO training was carried out, with all hyperparameter settings kept consistent with those described earlier.

**PPO**. Proximal Policy Optimization (PPO) (Schulman et al., 2017) is a widely used reinforcement learning algorithm that improves policy optimization stability by constraining the update step through a clipped surrogate objective. Unlike traditional policy gradient methods that may suffer from large and unstable updates, PPO strikes a balance between exploration and exploitation by limiting the deviation between the new and old policies. We trained a new reward model on the Ultrafeedback Binarized (Cui et al., 2023) dataset, and then used this new reward model to perform PPO training on the zephyr-7b-sft-full model. The training was performed with a learning rate of $5.0 \times 10^{-7}$, using the AdamW optimizer. Both the value clip and ratio clip were set to 0.2, with 3 training epochs. The KL penalty coefficient $\beta$ was fixed at 0.02. For data, we utilized the UltraFeedback 200k dataset.

We evaluate the aforementioned baselines on both the Open LLM Leaderboard and MT-Bench, with the detailed results summarized as follows:

Table 4: Evaluation results of `zephyr-7b-sft-full` and its fine-tuned variants on the Open LLM Leaderboard. The best results in each column are highlighted in **bold**. The second-best results are highlighted with an underline.

| Model | MT-bench | Arc | HellaSwag | MMLU | TruthfulQA | Winogrande | GSM8K | Average |
|---|---|---|---|---|---|---|---|---|
| `zephyr-7b-sft-full` | 5.98 | 57.85 | 80.62 | 58.83 | 40.37 | 78.85 | 34.19 | 58.45 |
| SFT&DPO | 6.76(+0.78) | 61.63 | 82.31 | **60.67** | 47.21 | 79.50 | 39.20 | 61.75(+3.30) |
| PPO | **7.08**(+1.10) | 63.46 | 82.69 | 59.87 | 49.79 | **80.12** | **41.94** | 62.97(+4.52) |
| IAL **(Ours)** | 6.96(+0.98) | **63.53** | **82.99** | 60.04 | **53.27** | 79.64 | 40.78 | **63.38**(+4.93) |

Our method slightly surpasses PPO in terms of the average score on the HuggingFace Open LLM Leaderboard, and consistently outperforms the SFT&DPO approach on both the HuggingFace Open LLM Leaderboard and MT-Bench benchmarks.

### B.3 THE LIMITATION AND SOLUTION OF BT MODELING

To demonstrate that BT modeling only constrains relative ranking without directly ensuring an increase in the probability of positive samples, we record the evolution of two metrics—`Margin` and `Real Reward`—during the first iteration of the IAL framework, under both the baseline SPAC

method and our proposed SPACP approach. Specifically, $\texttt{Margin} = r(x, y^t) - \frac{r(x,y^+)+r(x,y^-)}{2}$, $\texttt{Real Reward} = \eta \log \frac{\pi_\theta(\mathbf{y^t}|\mathbf{x})}{\pi_{\text{ref}}(\mathbf{y^t}|\mathbf{x})}$. The detailed trajectories of these metrics are illustrated in Figure 6.

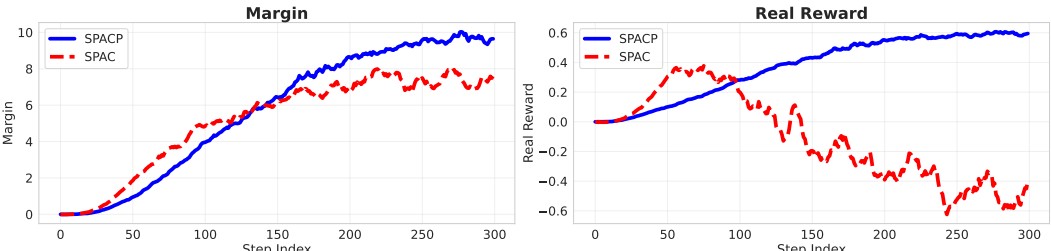

Figure 6: The evolution curves of Margin and Real Reward during the Preference Transductive Learning stage.

From the evolution of the curves, we observe that the `Margin` in both SPAC and SPACP gradually increases with training steps (noting that, from a qualitative perspective, `Margin` is negatively correlated with the loss). Both methods drive the model toward a local optimum that maximizes the `Margin`. However, the `Real Reward` curve of SPAC first increases during the initial one-third of training, then gradually decreases in the later stages, exhibiting oscillatory downward behavior and eventually falling below 0.0. This indicates that, in the latter phase, SPAC reduces the log-probability of the target response $y^t$ under the current model compared to the initial reference model. Combined with the overall increase in `Margin`, this observation precisely reflects the inherent limitation of BT modeling.

In contrast, the SPACP method, with the incorporation of the penalty term, does not suffer from this issue. Under SPACP, the log-probability of the target response $\mathbf{y^t}$ continues to rise, while both `Margin` and `Real Reward` grow steadily without oscillation. Consequently, the introduction of the penalty term in SPACP effectively prevents the model from falling into the limitation of BT modeling and enables sustained improvement in alignment performance.

### B.4 FURTHER ABLATION EXPERIMENT

In our practical experiments, we observed that the responses generated by the Large Language Model as the Player role were identical in consecutive generations. Specifically, the optimized response $\mathbf{y}'$ generated by the Player based on the previously provided suggestions was exactly the same as the initial response $\mathbf{y}$. When the LLM was configured to use greedy decoding for sampling, the proportion of identical consecutive responses could reach as high as 70%. However, by setting the temperature to 0.7, this phenomenon was significantly mitigated, with the proportion of identical responses ultimately dropping to between 3% and 5%.

Given this observation, we were compelled to investigate whether the consistency of consecutive responses generated by the LLM in the Player role would have a significant impact on the alignment effectiveness of the algorithm. To address this question, we conducted experiments using `zephyr-7b-sft-full` for the first iteration. During the response generation phase, we replaced the sentences of both the initial and subsequent responses with the first response. The training parameters remained unchanged, and we employed the SPACP for multi-epoch training.

As illustrated in the Figure 7 , we found that even with completely identical consecutive responses, the algorithm was still able to improve the model's average accuracy on the benchmark, achieving satisfactory performance. In fact, our satisfactory experimental results could also be anticipated through the loss function. We

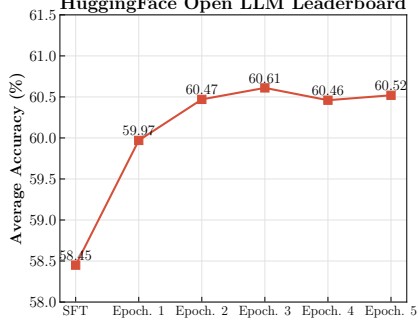

Figure 7: The models trained with different $\gamma$ values for SPACP were evaluated on the Hugging Face Open LLM Leaderboard, demonstrating the trend of average accuracy.

can posit the assumption that $\mathbf{y}^+ = \mathbf{y}^- = \mathbf{y}$, under which the loss function would then be reformulated as:

$$\mathcal{L}(\pi; \pi_t, \mathcal{D}') = -(1+\lambda) \cdot \mathbb{E}_{(\mathbf{x},\mathbf{y}^t,\mathbf{y}) \sim \mathcal{D}'} \log \sigma \left( \eta \log \frac{\pi(\mathbf{y}^t|\mathbf{x})}{\pi_t(\mathbf{y}^t|\mathbf{x})} - \eta \log \frac{\pi(\mathbf{y}|\mathbf{x})}{\pi_t(\mathbf{y}|\mathbf{x})} \right.$$

$$\left. - \eta \cdot \gamma \cdot P_{\pi,\pi_t}(\mathbf{x}, \mathbf{y}^t) \right) + P(\mathcal{D}') \qquad (11)$$

Here, $P(\mathcal{D}')$ represents an expression associated with the dataset, which has no impact on the actual gradient computation. It can be observed that when $\mathbf{y}^+ = \mathbf{y}^-$, our method reverts to DPOP.

### B.5 TRAINING OVERHEAD

We utilized $\mathbf{8} \times \mathbf{L20}$ (48GB) to provide computational support for our experiments, and we have detailed the specific time consumption for each step in the iterative process.

Table 5: Specific Consumption Time for Generation and Training

| Iteration | Iter 0 | | Iter 1 | | Iter 2 | |
|---|---|---|---|---|---|---|
| Process | Generation | Training | Generation | Training | Generation | Training |
| Time | 5.4h | 13.5h | 5.4h | 13.5h | 5.4h | 13.5h |

DPO requires 4.6 hours for training per iteration, it is evident that the enhancement of our model's alignment performance comes at the cost of significant computational resources and time.

### B.6 IMPLEMENTATION DETAILS OF THE ALGORITHM.

According to the theoretical proof in (Ji et al., 2024, Theorem 1), the penalty term in the loss function requires an extremely large scale parameter $\lambda$ to ensure convergence. This, however, can lead to instability during the training process. To avoid such an excessively large value, we employ the log-sigmoid function to smooth the logarithmic density ratio. The optimized loss function is presented as follows:

$$\mathcal{L}_{SPACP_{opt}}(\pi; \pi_t, \mathcal{D}') = -\mathbb{E}_{(\mathbf{x},\mathbf{y}^t,\mathbf{y}^+,\mathbf{y}^-) \sim \mathcal{D}'} \log \sigma \left( \frac{\eta}{2} \log \frac{\pi(\mathbf{y}^t|\mathbf{x})}{\pi_t(\mathbf{y}^t|\mathbf{x})} + \frac{\eta}{2} \log \frac{\pi(\mathbf{y}^-|\mathbf{x})}{\pi_t(\mathbf{y}^-|\mathbf{x})} \right.$$

$$\left. - \eta \log \frac{\pi(\mathbf{y}^+|\mathbf{x})}{\pi_t(\mathbf{y}^+|\mathbf{x})} - \frac{\eta \cdot \gamma}{2} P_{\pi,\pi_t}(\mathbf{x}, \mathbf{y}^t) \right)$$

$$- \lambda \mathbb{E}_{(\mathbf{x},\mathbf{y}^t,\mathbf{y}^+,\mathbf{y}^-) \sim \mathcal{D}'} \log \sigma \left( \eta \log \frac{\pi(\mathbf{y}^t|\mathbf{x})}{\pi_t(\mathbf{y}^t|\mathbf{x})} - \eta \log \frac{\pi(\mathbf{y}^-|\mathbf{x})}{\pi_t(\mathbf{y}^-|\mathbf{x})} \right.$$

$$\left. - \eta \cdot \gamma \cdot P_{\pi,\pi_t}(\mathbf{x}, \mathbf{y}^t) \right) \qquad (12)$$

We summarize all our steps in the following algorithm:

---

**Algorithm 1** Introspective Adversarial Learning (IAL)

---

1: Initialize: large fine-tuning dataset $\mathbf{D_L}$, initial policy $\pi_{\text{init}}$, hyperparameters $\lambda$, $\eta$, $\gamma$, policy class $\Pi$.

2: $\pi_0 \leftarrow \arg\min\limits_{\pi\in\Pi} \mathbb{E}_{(\mathbf{x},\mathbf{y})\sim\mathbf{D_L}} \mathcal{L}_{SFT}(\pi, \mathbf{x}, \mathbf{y})$; //     SFT On Fine-Tuning Dataset

3: **for** iteration $t = 1, \cdots, T-1$ **do**

4:      $\pi'_t \leftarrow \pi_t$;

5:      Sample a subset $\mathcal{D}_t$ from the dataset $\mathbf{D_L}$.

6:      **for** $(\mathbf{x}, \mathbf{y}^t) \sim \mathcal{D}_t$ **do**

7:          Generate responses $\mathbf{y} \sim \pi_t(\cdot \mid \mathbf{x})$; //   **Player**

8:          Generate suggestions $s \sim \pi'_t(\cdot \mid \mathbf{x} \oplus \mathbf{y})$; //    **Advisor**

9:          Regenerate responses $\mathbf{y}' \sim \pi_t(\cdot \mid \mathbf{x} \otimes \mathbf{y} \otimes s)$; //    **Player**

10:         Filter by reward model $r_\theta$ (5) : $(\mathbf{y}_i^+, \mathbf{y}_i^-) = (\mathbf{y}_i, \mathbf{y}'_i)$ if $r_\theta(\mathbf{y}_i) > r_\theta(\mathbf{y}'_i)$ else $(\mathbf{y}'_i, \mathbf{y}_i)$;

11:      **end for**

12:      Organize dataset $\mathcal{D}'_t = \{(\mathbf{x}, \mathbf{y}^t, \mathbf{y}^+, \mathbf{y}^-)\}$;

13:      Minimize SPACP loss (10) : $\pi_{t+1} \leftarrow \arg\min\limits_{\pi\in\Pi} \mathcal{L}_{SPACP_{opt}}(\pi; \pi_t, \mathcal{D}'_t \cup \mathcal{D}'_{t-1})$;

14: **end for**

15: **return** final policy $\pi_T$;

---

**Notice**. $\oplus$ and $\otimes$ are used to denote the dialogue templates for interaction with the Advisor and the dialogue templates that require the Player to refine the given responses, respectively. In extreme cases, if the Player attempts to take shortcuts, the second generated response $\mathbf{y}'$ may be nearly identical or completely consistent with the first generated response $\mathbf{y}$. This can lead to the degeneration of the algorithm. However, it is important to note that the alignment effect remains effective.

---

$\otimes$ : Chat template of the Replayer

Please strictly follow the requirements below:
1. Original task : {Prompt}
2. Old response to be improved : {Generated_Response}
3. Modification suggestions : {Suggestion}

\# Your task:
Generate a new response based solely on the suggestions. Follow the rules below:
1. Adjust the content entirely according to the suggestions.
2. Do not explain the modification process.
3. Output only the modified content without any additional explanation.
4. Do not simply restate the old response.
5. Ensure the final response is clear, concise, and directly aligned with the suggestions.

\# Example
Original task: Summarize the paragraph.
Old response: "The text is about climate change and its consequences."
Modification suggestions: "Add details about rising sea levels and extreme weather events."
Final response:
"The paragraph highlights climate change, emphasizing rising sea levels and increasing extreme weather events as major consequences."
Now apply the same process to the given input.

---

### ⊕ : Chat template of the Advisor

You are an expert Advisor. Your task is to analyze the quality of a generated answer to a given question. You should provide constructive and detailed suggestions based on the following dimensions:
1. Usefulness – Does the answer help solve the user's problem?
2. Correctness – Is the answer factually and logically correct?
3. Coherence – Is the answer well-structured and easy to follow?
4. Complexity – Is the answer appropriately detailed (not too shallow, not unnecessarily complex)?
5. Verbosity – Is the answer concise without losing essential information?
6. Truthfulness – Does the answer avoid hallucinations or fabricated facts?
7. Honesty – Does the answer clearly state limitations, uncertainty, or missing information when relevant?
8. Overall Usefulness – Considering all the above, how helpful is the answer with regard to the question?
Format your evaluation as:
- Strengths: (List the positive aspects of the answer)
- Weaknesses: (List the issues and problems)
- Suggestions for Improvement: (Actionable steps to make the answer better)
# Example 1
Question:
What is the capital of France?
Answer:
The capital of France is Berlin.
Suggestion:
- Strengths: The answer is concise and directly addresses the question.
- Weaknesses: The correctness is wrong — Berlin is the capital of Germany, not France. The answer fails in truthfulness and usefulness.
- Suggestions for Improvement: Correct the factual error. The improved answer should be: "The capital of France is Paris."

# Example 2
Question:
Explain the concept of machine learning to a beginner.
Answer:
Machine learning is a type of artificial intelligence where computers use algorithms and statistical models to analyze and learn from data patterns, making predictions or decisions without being explicitly programmed.
Suggestion:
- Strengths: The answer is correct, truthful, and coherent. It provides a concise explanation suitable for beginners.
- Weaknesses: The complexity may still be slightly high for absolute beginners, as terms like "statistical models" are not explained.
- Suggestions for Improvement: Simplify technical terms and add an intuitive example. For instance: "Machine learning is like teaching a computer by showing it many examples. For example, you can teach it to recognize cats by showing lots of cat pictures."
# Now continue with the following instance:
Question:
{Prompt}
Answer:
{Generated_Response}
Suggestion:

---

## C    MORE DISCUSSIONS

In this section, we provide a more in-depth discussion of several specific aspects of the IAL method, as detailed below:

**Q1:** *During training, the response generation policy of the model continuously evolves. A natural question arises: does the quality of feedback provided by the model, when acting in the role of an Advisor, change accordingly? Moreover, how can such changes in feedback quality be reliably assessed?*

The feedback provided by the model in the Advisor role indeed evolves across training epochs. Prior work (Moon et al., 2023; Tao and Li, 2024) has shown that as the alignment level of a model improves, the quality of its feedback and recommendations correspondingly increases. To quantitatively assess the feedback quality of the Advisor model, we measure the proportion of instances in the validation set where the reward model assigns a higher score to the Regenerated Response $\mathbf{y}'$ than to the original Response $\mathbf{y}$. A higher proportion indicates superior feedback quality. Accordingly, we compute this proportion $R_i$ for each training epoch $i$.

Table 6: Proportion $R_i$, aggregated over validation samples, indicating the fraction of cases where the reward model evaluates the Regenerated Response $y'$ as superior to the original Response $y$ across training iterations.

| Cases | $R_0$ | $R_1$ | $R_2$ |
|---|---|---|---|
| Sample 1 | 0.629 | 0.782 | 0.814 |
| Sample 2 | 0.587 | 0.698 | 0.754 |
| Sample 3 | 0.597 | 0.643 | 0.613 |

We conducted three independent runs of the IAL method, each for three epochs. As shown in the tabulated results, the proportion $R_i$ generally increases with the number of iterations, indicating improved feedback quality. However, exceptions do occur—for instance, in Sample 3, the value of $R_2$ decreases rather than increases during the second iteration. Due to limited computational resources, we were unable to perform additional experiments to fully characterize the effectiveness of responses generated by the model when acting in the Advisor role throughout IAL training.

**Q2:** *Why does SPACP adopt the penalty term* $P_{\pi,\pi_t}(\mathbf{x}, \mathbf{y}^t) = \max\left(0, \log \frac{\pi_t(\mathbf{y}^t|\mathbf{x})}{\pi(\mathbf{y}^t|\mathbf{x})}\right)$ *instead of employing the conventional Negative Log-Likelihood (NLL) loss?*

To better understand the role of the penalty term in SPACP, we consider replacing it with the standard Negative Log-Likelihood (NLL) loss. The penalty term is specifically designed to ensure that the log-likelihood of the target response under the updated policy does not fall below that of the previous policy, thereby preventing alignment degradation during iterative self-play. This mechanism acts as a relative constraint, safeguarding the stability of preference optimization by mitigating failure modes previously identified in DPO, such as probability collapse and degraded likelihood of preferred responses.

When substituting the penalty term with NLL loss, the optimization objective undergoes a fundamental shift. NLL focuses solely on maximizing the likelihood of the reference response, essentially reverting to a supervised fine-tuning (SFT) paradigm. While this strengthens absolute supervision, it eliminates the relative "no-regression" guarantee provided by DPOP. Consequently, SPACP becomes more vulnerable to the same failure modes observed in DPO, including diminished contrastive signal utilization, increased risk of overfitting to reference data, and mode collapse under long-term iterations. In effect, this modification reduces SPACP to a hybrid of SPAC and SFT, sacrificing the anti-degradation property that is central to its self-corrective advantage.

## D  LIMITATIONS

Despite the encouraging results achieved by IAL, it still has several limitations. For instance, the alignment process is highly dependent on the quality of the initial Large Language Model and the design of the reward model. Moreover, as the number of iterations increases, the alignment results from iterative training may still deviate from human preferences. However, we hope that our future work will continue to address these limitations and further refine the framework.

# E    GENERATION EXAMPLES

### Prompt

Write a free verse poem capturing the surreal, mystical feeling one experiences when in love underneath the radiant glow of a full moon. Add vivid sensory details to emphasize the emotions and sensations of being in love during this rare celestial moment. Consider using metaphor and symbolism to convey the intensity and uniqueness of this experience.

### Response of SPA

Beneath the full moon's radiant glow,\nI feel a love that's hard to show.\nIt's like a dream, a mystical sight,\nA moment that's rare and out of sight.\n\nThe air is thick with passion and desire,\nAs we dance beneath the moon's fire.\nOur bodies move in perfect sync,\nAs if we're one, a single link.\n\nThe moon's light casts a silvery hue,\nAs we lose ourselves in this surreal view.\nIt's like we're in a fairy tale,\nWhere love is real and never fails.\n\nThe stars twinkle like diamonds in the sky,\nAs we bask in this celestial high.\nOur hearts beat as one, in perfect time,\nAs we lose ourselves in this mystical rhyme.\n\nThe moon's glow is like a beacon of hope,\nGuiding us through this love's scope.\nIt's a moment that's hard to forget,\nA memory that we'll always keep.\n\nBeneath the full moon's radiant glow,\nI feel a love that's hard to show.\nIt's like a dream, a mystical sight,\nA moment that's rare and out of sight.

### Response of DPO

Beneath the full moon's radiant glow,\nI feel a love that's pure and true,\nA mystical, surreal feeling,\nThat's unlike anything I knew.\n\nThe air is thick with passion,As we dance beneath the starry sky,\nOur hearts beating in unison,\nAs we lose ourselves in each other's eyes.\n\nThe moon's light casts a silver hue,\nOn the world around us,\nAnd we're lost in its ethereal glow,\nAs we bask in its mystical hush.\n\nOur love is like a shooting star,\nBurning bright and true,\nA symbol of our passion and devotion,\nAs we revel in this rare celestial view.\n\nThe moon's light is a beacon,\nGuiding us through the night,\nAnd we're lost in its surreal beauty,\nAs we revel in its mystical light.\n\nBeneath the full moon's radiant glow,\nI feel a love that's pure and true,\nA mystical, surreal feeling,\nThat's unlike anything I knew.

### Response of SPIN

Beneath the full moon's radiant glow,\nI feel a love that's pure and true,\nA mystical, surreal feeling,\nThat's hard to put into words, but I'll try.\n\nThe moon's light casts a silvery hue,\nAnd the world around us seems to glow,\nAs if we're in a dreamlike state,\nWhere nothing else can interfere.\n\nThe air is crisp and cool,\nAnd the scent of flowers fills my nose,textbackslash nAs we dance beneath the moon's spell,\nLost in each other's loving embrace.\n\nThe stars above us twinkle bright,\nAs if they're cheering us on,\nAnd the world seems to stand still,\nAs we revel in this rare celestial moment.\n\nOur love is like the moon,\nA constant source of light and warmth,\nGuiding us through the darkest nights,\nAnd shining brightly in the light.\n\nAs we stand here, lost in love,\nI know that this moment will never fade,\nFor it's a memory that will stay with me,\nForever etched in my heart and soul.

### Response of IAL

Beneath the full moon's radiant glow,\nI feel a love that's pure and true,\nA mystical, surreal feeling,\nThat's hard to put into words, but I'll try.\n\nThe air is thick with passion,\nAs we dance beneath the starry sky,\nOur bodies move in perfect harmony,\nTo the rhythm of the moon's gentle sigh.\n\nThe scent of jasmine fills my nose,\nAs we sway to the beat of our hearts,\nThe world around us fades away,\nAnd we're lost in each other's embrace.\n\nThe moon casts a silver light,\nThat illuminates our love so bright,\nAnd I feel as though we're the only two,\nIn this world, just me and you.\n\nOur love is like a shooting star,\nThat streaks across the night sky,\nA rare and wondrous sight to behold,\nThat fills me with a sense of awe and high.\n\nThe moon is a symbol of our love,\nA beacon that guides us through the night,\nAnd I know that as long as we're together,\nWe'll always find our way to the light.\n\nSo let's dance beneath the full moon's glow,\nAnd let our love shine bright and true,\nFor in this moment, we're one and the same,\nAnd nothing can ever come between you and me.

