# OpenReview forum: "Introspective Adversarial Learning: Autonomous and Continual Preference Learning for LLM Alignment"
_ICLR.cc/2026/Conference — ICLR 2026 Conference Withdrawn Submission_

### Official Review · Reviewer_Ddec · 2025-10-16

**Soundness:** 1
**Presentation:** 2
**Contribution:** 1
**Rating:** 0
**Confidence:** 5

**Summary:**

The paper proposes Introspective Adversarial Learning (IAL), a self-improvement alignment framework where a Player LLM generates an initial response, an Advisor LLM provides targeted suggestions, and the Player regenerates an improved response; a reward model then ranks the pair to form preference data for offline preference optimization via a SPAC-style objective with a positive-preserving modification (SPACP). Experiments on zephyr‑7B‑sft‑full with UltraChat/UltraFeedback report gains over DPO, SPA, and SPIN on the Open LLM Leaderboard (v1) and MT‑Bench, with an ablation on the penalty hyperparameter and a brief appendix result on Qwen

**Strengths:**

Clear decomposition into self-refinement data generation and preference optimization, which mirrors practical workflows and is easy to reproduce.

Uses an explicit “positive-preserving” modification to address BT-style relative-only issues in DPO/SPAC, with an ablation on γ.

Iterative pipeline and reporting across multiple iterations help illustrate convergence dynamics rather than single-shot gains.

**Weaknesses:**

Backbone choice and recency: Using zephyr‑7B‑sft‑full as the main backbone feels dated and weakens external validity; this model was popular circa 2023. At minimum, please include strong, current 7B baselines such as Qwen‑2.5‑Instruct‑7B and Llama‑3.1/3.2‑Instruct‑8B with matched training setups. The appendix reports “Qwen‑2.5‑3B,” but it’s unclear whether this is the base, instruct, or SFT variant—this should be clarified and expanded to 7B.

Leaderboard versioning and setup: The Open LLM Leaderboard evaluation appears to follow the older v1 style and a SPIN-like setup; more convincing evidence would use the updated v2 protocols and clearly document eval agents, few-shot regimes, and any normalization or debiasing steps.

MT‑Bench limitations: MT‑Bench is known to be noisy and less discriminative today; Arena‑Hard (v2) and AlpacaEval 2.0 (length-controlled) are more informative for chat/instruction-following, and math verification should include recent suites such as Math500 and AIME’24/’25-style evaluations.

Marginal gains vs. complexity: The overall framework adds multiple moving parts (Player/Advisor prompting, PairRM curation, SPACP) yet the reported improvements over SPIN/DPO are modest; please quantify cost/benefit (GPU-hours, RM queries, data generation volume) and show statistically significant margins across multiple seeds and models.

Novelty positioning: The contribution reads as a composition of existing ideas (self-refinement/self-play, adversarial critic, positive-preserving objectives) applied on older models/benchmarks. Please sharpen what is new at the algorithmic level beyond SPIN/SPAC/DPO variants, provide theoretical intuition or guarantees for SPACP, and demonstrate unique empirical advantages on modern backbones and benchmarks.

Reward model dependence and circularity: Reliance on PairRM as judge risks overfitting to a particular evaluator. Include cross-judge robustness (e.g., different RMs/LLM-as-judge, human evaluation, blinded pairwise) and measure label quality drift across iterations to rule out circular artifacts.

Clarity on data flow: UltraChat/UltraFeedback usage, data mixing across iterations, and de-duplication policies need clearer documentation to ensure no leakage or inadvertent contamination, especially when comparing across methods.

**Questions:**

See Weaknesses section.

---

### Official Review · Reviewer_pPV8 · 2025-11-01

**Soundness:** 2
**Presentation:** 3
**Contribution:** 2
**Rating:** 4
**Confidence:** 3

**Summary:**

The paper set up a Player-Advisor loop: the Player answers, the Advisor suggests edits, the Player rewrites, and a reward model (PairRM) selects the better of the two; those pairs train the policy with a SPAC-like objective plus a new positive term (SPACP). Trained on UltraChat-derived prompts with zephyr-7b-sft-full, they iterate this process and report consistent gains on the Open LLM Leaderboard and MT-Bench vs. DPO/SPIN/SPA.

**Strengths:**

+ Clear, reproducible pipeline with concrete steps and illustration.
+ Results look uniformly better than SFT&DPO and broadly competitive with PPO on the HF leaderboard and MT-Bench.
+ The SPACP tweak directly targets a known BT/DPO issue (only pushing down the “bad”) and is well-justified.

**Weaknesses:**

- The performance of the method seems heavily dependent on the reward model (PairRM). There is no guarantee of the RM's performance on a wider range of tasks or domain-specific tasks. Figure 4 (left) also confirms this concern.
- The advisor's contribution is not isolated. The paper shows many cases where the second response is identical under greedy decoding, yet training still helps: this suggests gains may come from the objective, not the advice. An “advisor off / random second rollout” ablation is missing.
- DPO baseline mismatch. The paper compares to DPO trained on UltraFeedback, not DPO trained on the generated PairRM-labeled pairs. It is hard to tell whether the win is from objective vs. data.
- Claimed gains are modest and somewhat mixed across tasks. PPO is close on averages in places.

**Questions:**

- How sensitive is IAL to the choice of reward model? How robust are the results when swapping PairRM for a different RM? Does RM adaptation materially help, or does IAL reduce the need for RM specialization?
- Advisor ablation: Compare (a) advised second generation, and (b) random/independent second rollout. If the preference is determined by external RM anyway, what is the actual delta the advisor introduced?
- What is the quantitative inference overhead introduced by the player-advisor model. How is it justified by its isolated gain?

---

### Official Review · Reviewer_muBz · 2025-11-01

**Soundness:** 2
**Presentation:** 2
**Contribution:** 2
**Rating:** 2
**Confidence:** 4

**Summary:**

This paper proposes a new preference learning framework, Introspective Adversarial Learning (IAL), where the model alternates between Player and Advisor roles to generate self-refined preference data via the external reward model PairRM, thereby achieving alignment without human feedback.

**Strengths:**

1.Novel self-play formulation:The Player–Advisor mechanism is conceptually elegant and extends traditional self-play alignment with an introspective adversarial loop.

2.Practical efficiency:Achieves alignment improvement without costly human preference annotation while maintaining strong general task performance.

3.Comprehensive evaluation:Consistent superiority over DPO, SPA, and SPIN on MT-Bench and the Open LLM Leaderboard.

**Weaknesses:**

1.Dependence on external models:The reliance on PairRM for preference judgment weakens the claim of full autonomy.

2.Limited cost analysis:While human supervision is removed, the paper does not adequately discuss the computational and time overhead introduced by iterative generation and ranking, nor compare it with DPO or other baselines.

3.Lack of introspective analysis – The paper lacks ablation or theoretical analysis on the stability, convergence, and potential failure modes of the introspective adversarial mechanism.

**Questions:**

1.Since PairRM is used to determine preference data, the accuracy and bias of this reward model will directly affect the alignment quality. Have you analyzed its sensitivity?

2.The method introduces substantial computational overhead for modest gains over DPO. It would be valuable to compare with GRPO or other recent reinforcement-style alignment methods to justify efficiency.

3.The PPO comparison appears unfair:PPO uses a separately trained reward model on UltraFeedback, while IAL leverages PairRM. The comparison should ideally use the same reward model for consistency.

---

### Official Review · Reviewer_yEmh · 2025-11-02

**Soundness:** 2
**Presentation:** 1
**Contribution:** 2
**Rating:** 2
**Confidence:** 4

**Summary:**

This paper introduces a Player-Advisor mechanism to collect preference pairs, where the first response comes directly from the policy model and the second one is the refined response based on feedback from the model itself. The authors further propose a new learning algorithm, Self-Play with Adversarial Critic-Positive (SPACP), which includes a penalty to encourage the model to retain high probability on the human annotated responses. Experiments show better performance than other baselines.

**Strengths:**

This paper adopts refined responses based on feedback from the model itself to get better preference pairs. The self-improvement paradigm could be interesting to the community.

**Weaknesses:**

* Ablations are not convincing enough to justify the proposed method.
* The equations are not always clear.
* The use of human response demands human annotation and makes the learning off-policy, which may hurt performance.

**Questions:**

* The formulation is confusing, which reduces the readability. In Equation 8, where is the log \sigma in the second term from? In Equation 6 and 8, \lambda is used for the penalty term, but in Equation 9 and 10, it changed to the preference learning term.
* In lines 283-284, it seems that the purpose of SPACP is to retain the log-likelihood of y^+, but it’s actually used y^t. Why use the human annotation here, which clearly demands more human involvement? Also, the placement of the penalty term related to y^t lacks empirical justification? what if just using y^+ instead (note y^+ is from the model, while y^t is totally off-policy)?
* Would the off-policy learning caused by y^t hurt model performance? Please add a discussion.
* There is no clear comparison between SPAC and SPACP on the final model performance. It’s unclear how serious the motivation of SPACP is in practice.
* The paper claims “reducing alignment costs”, but it’s not always clear how the proposed method compares to the others on the cost, particularly the human efforts. Please add a table detailing it for different training stages.

---

### Note · Authors · 2025-11-28

I have read and agree with the venue's withdrawal policy on behalf of myself and my co-authors.